# Neural Guided Constraint Logic Programming for Program Synthesis

**Lisa Zhang[1,2], Gregory Rosenblatt[4], Ethan Fetaya[1,2], Renjie Liao[1,2,3], William E. Byrd[4], Matthew Might[4], Raquel Urtasun[1,2,3], Richard Zemel[1,2]**

[1]University of Toronto, [2]Vector Institute, [3]Uber ATG, [4]University of Alabama at Birmingham
[1]{lczhang,ethanf,rjliao,urtasun,zemel}@cs.toronto.edu
[4]{gregr,webyrd,might}@uab.edu

## Abstract

Synthesizing programs using example input/outputs is a classic problem in artificial intelligence. We present a method for solving Programming By Example (PBE) problems by using a neural model to guide the search of a constraint logic programming system called miniKanren. Crucially, the neural model uses miniKanren's internal representation as input; miniKanren represents a PBE problem as recursive constraints imposed by the provided examples. We explore Recurrent Neural Network and Graph Neural Network models. We contribute a modified miniKanren, drivable by an external agent, available at https://github.com/xuexue/neuralkanren. We show that our neural-guided approach using constraints can synthesize programs faster in many cases, and importantly, can generalize to larger problems.

## 1 Introduction

Program synthesis is a classic area of artificial intelligence that has captured the imagination of many computer scientists. Programming by Example (PBE) is one way to formulate program synthesis problems, where example input/output pairs specify a target program. In a sense, supervised learning can be considered program synthesis, but supervised learning via successful models like deep neural networks famously lacks interpretability. The clear interpretability of programs as code means that synthesized results can be compared, optimized, translated, and proved correct. The manipulability of code makes program synthesis continue to be relevant today.

Current state-of-the-art approaches use symbolic techniques developed by the programming languages community. These methods use rule-based, exhaustive search, often manually optimized by human experts. While these techniques excel for small problems, they tend not to scale. Recent works by the machine learning community explore a variety of statistical methods to solve PBE problems more quickly. Works generally fall under three categories: differentiable programming [1, 2, 3], direct synthesis [4, 5], and neural guided search [6, 7].

This work falls under neural guided search, where the machine learning model guides a symbolic search. We take integrating with a symbolic system further: we use its internal representation as input to the neural model. The symbolic system we use is a constraint logic programming system called miniKanren[1][8], chosen for its ability to encode synthesis problems that are difficult to express in other systems.

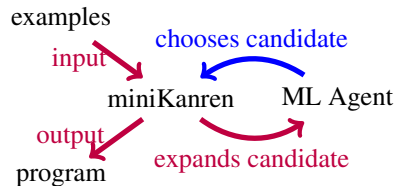

Figure 1: Neural Guided Synthesis Approach

Specifically, miniKanren does not rely on types, is able to to complete partially specified programs, and has a straightforward implementation [9]. miniKanren searches for a candidate program that satisfies the recursive constraints imposed by the input/output examples. Our model uses these constraints to score candidate programs and guide miniKanren's search.

Neural guided search using constraints is promising for several reasons. First, while symbolic approaches outperform statistical methods, they have not demonstrated an ability to scale to larger problems; neural guidance may help navigate the exponentially growing search space. Second, symbolic systems exploit the compositionality of synthesis problems: miniKanren's constraints select portions of the input/output examples relevant to a subproblem, akin to having a symbolic attention mechanism. Third, constraint lengths are relatively stable even as we synthesize more complex programs; our approach should be able to generalize to programs larger than those seen in training.

To summarize, we contribute a novel form of neural guided synthesis, where we use a symbolic system's internal representations to solve an auxiliary problem of constraint scoring using neural embeddings. We explore two models for scoring constraints: Recurrent Neural Network (RNN) and Graph Neural Network (GNN) [10]. We also present a "transparent" version of miniKanren with visibility into its internal constraints, available at https://github.com/xuexue/neuralkanren.

Our experiments focus on synthesizing programs in a subset of Lisp, and show that scoring constraints help. More importantly, we test the generalizability of our approach on three families of synthesis problems. We compare against state-of-the-art systems $\lambda^2$ [11], Escher [12], Myth [13], and RobustFill [4]. We show that our approach has the potential to generalize to larger problems.

## 2 Related Work

Programming by example (PBE) problems have a long history dating to the 1970's [14, 15]. Along the lines of early works in program synthesis, the programming languages community developed search techniques that enumerate possible programs, with pruning strategies based on types, consistency, and logical reasoning to improve the search. Several state-of-the-art methods are described in Table 1.

Table 1: Symbolic Methods

| Method | Direction | Search Strategy | Type Discipline |
|---|---|---|---|
| miniKanren [8, 16] | Top-down | Biased-Interleaving | Dynamic |
| $\lambda^2$ [11] | Top-down | Template Complexity | Static |
| Escher [12] | Bottom-up | Forward Search / Conditional Inference | Static |
| Myth [13] | Top-down | Iterative Deepening | Static |

The method $\lambda^2$ [11] is most similar to miniKanren, but specializes in numeric, statically-typed inputs and outputs. Escher [12] is built as an active learner, and relies on the presence of an oracle to supply outputs for new inputs that it chooses. Myth [13] searches for the smallest program satisfying a set of examples, and guarantees parsimony. These methods all use functional languages based on the $\lambda$-calculus as their target language, and aim to synthesize general, recursive functions.

Contributions by the machine learning community have grown in the last few years. Interestingly, while PBE problems can be thought of as a meta-learning problem, few works explore this relationship. Each synthesis problem can be thought of as a learning problem [17], so learning the synthesizer can be thought of as meta-learning. Instead, works generally fall under direct synthesis, differentiable programming, and neural guided synthesis.

**Direct Synthesis** In direct synthesis, the program is produced directly as a sequence or tree. One domain where this has been successful is string manipulation as applied to spreadsheet completion, as in FlashFill [18] and its descendants [5, 4, 19]. FlashFill [18] uses a combination of search and carefully crafted heuristics. Later works like [5] introduce a "Recursive-Reverse-Recursive Neural Network" to generate a program tree conditioned on input/output embeddings. More recently, RobustFill [4] uses bi-directional Long Short-Term Memory (LSTM) with attention, to generate programs as sequences. Despite flattening the tree structure, RobustFill achieved much better results (92% vs 38%) on the FlashFill benchmark. While these approaches succeed in the practical domain of string manipulation, we are interested in exploring manipulations of richer data structures.

**Differentiable Programming**   Differentiable programming involves building a differentiable interpreter, then backpropagating through the interpreter to learn a *latent* program. The goal is to infer correct outputs for new inputs. Work in differentiable programming began with the Neural Turing Machine [3], a neural architecture that augments neural networks with external memory and attention. Neural Programmer [1] and Neural Programmer-Interpreter [2] extend the work with reusable operations, and build programs compositionally. While differentiable approaches are appealing, [20] showed that this approach still underperforms discrete search-based techniques.

**Neural Guided Search**   A recent line of work uses statistical techniques to guide a discrete search. For example, DeepCoder [6] uses an encoding of the input/output examples to predict functions that are likely to appear in the program, to prioritize programs containing those functions. More recently, [7] uses an LSTM to guide the symbolic search system PROSE (Microsoft Program Synthesis using Examples). The search uses a "branch and bound" technique. The neural model learns the choices that maximize the bounding function $h$ introduced in [18] and used for FlashFill problems. These approaches attempt to be search system agnostic, whereas we integrate deeply with one symbolic approach, taking advantage of its internal representation and compositional reasoning.

Other work in related domains shares similarities with our contribution. For example, [21] uses constraint-based solver to sample terms in order to complete a program sketch, but is not concerned with synthesizing entire programs. Further, [22] implements differentiable logic programming to do fuzzy reasoning and induce soft inference rules. They use Prolog's depth-first search as-is and learn constraint validation (approximate unification), whereas we learn the search strategy and use miniKanren's constraint validation as-is.

# 3   Constraint Logic Programming with miniKanren

This section describes the constraint logic programming language miniKanren and its use for program synthesis. Figure 1 summarizes the relationship between miniKanren and the neural agent.

## 3.1   Background

The constraint logic programming language miniKanren uses the relational programming paradigm, where programmers write *relations* instead of functions. Relations are a generalization of functions: a function $f$ with $n$ parameters can be expressed as a relation $R$ with $n + 1$ parameters, e.g., $(f\ x) = y$ implies $(R\ x\ y)$. The notation $(R\ x\ y)$ means that $x$ and $y$ are related by $R$.

In miniKanren queries, data flow is not directionally biased: any input to a relation can be unknown. For example, a query $(R\ \boxed{X}\ y)$ where $y$ is known and $\boxed{X}$ is an unknown, called a *logic variable*, finds values X where X and $y$ are related by $R$. In other words, given $R$ and $f$ defined as before, the query finds inputs X to $f$ such that $(f\ X) = y$. This property allows the relational translation of a function to run computations in reverse [16]. We refer to such uses of relations containing logic variables as *constraints*.

In this work, we are interested in using a relational form `evalo` of an interpreter EVAL to perform program synthesis[2]. In the functional computation (EVAL P I) = O, program P and input I are known, and the output O is the result to be computed. The same computation can be expressed relationally with (`evalo` P I $\boxed{O}$) where P and I are known and $\boxed{O}$ is an unknown. We can also synthesize programs from inputs and outputs, expressed relationally with (`evalo` $\boxed{P}$ I O) where $\boxed{P}$ is unknown while I and O are known. While ordinary evaluation is deterministic, there may be many valid programs P for any pair of I and O. Multiple uses of `evalo`, involving the same $\boxed{P}$ but different pairs I and O can be combined in a conjunction, further constraining $\boxed{P}$. This is how PBE tasks are encoded using an implementation of `evalo` for the target synthesis language.

```
(evalo P̲ I O)
  ⇒DISJ → (evalo (quote A̲) I O)
        → (evalo (car B̲) I O)
        → (evalo (cdr C̲) I O)
        → (evalo (cons D̲ E̲) I O)
        → (evalo (var F̲) I O)
           ...
```

Figure 2: Expansion of an `evalo` constraint

A miniKanren program internally represents a query as a constraint tree built out of conjunctions, disjunctions, and calls to relations (constraints). A relation like `evalo` is recursive, that is, defined in terms of invocations of other constraints including itself. Search involves unfolding a recursive constraint by replacing the constraint with its definition in terms of other constraints. For example, in a Lisp interpreter, a program P can be a constant, a function call, or another expression. Unfolding reveals these possibilities as clauses of a disjunction that replaces `evalo`. Figure 2 shows a partial unfolding of (`evalo` $\boxed{P}$ I O).

As we unfold more nodes, branches of the constraint tree constrain $\boxed{P}$ to be more specific. We call a partial specification of $\boxed{P}$ as a "candidate" partial program. If at some point we find a fully specified P that satisfies all relevant constraints, then P is a solution to the PBE problem.

In Figure 3, we show portions of the constraint tree representing a PBE problem with two input/output pairs. Each of the gray boxes corresponds to a separate disjunct in the constraint tree, representing a candidate. Each disjunct is a conjunction of constraints, shown one on each line. A candidate is viable only if the entire conjunction can be satisfied. In the left column (a) certain "obviously" failing candidates like (`quote` $\boxed{M}$) are omitted from consideration. The right column (c) also shows the unfolding of the selected disjunct for (`cons` $\boxed{D}$ $\boxed{E}$), where $\boxed{D}$ is replaced by its possible values.

By default, miniKanren uses a biased interleaving search [16], alternating between disjuncts to unfold. The alternation is "biased" towards disjuncts that have more of their constraints already satisfied. This search is *complete*: if a solution exists, it will eventually be found, time and memory permitting.

## 3.2 Transparent constraint representation

Typical implementations of miniKanren represent constraint trees as "goals" [16] built from opaque, suspended computations. These suspensions entangle both constraint simplification and the implicit search policy, making it difficult to inspect a constraint tree and experiment with alternative search policies.

One of our contributions is a miniKanren implementation that represents the constraint tree as a transparent data structure. It provides an interface for choosing the next disjunct to unfold, making it possible to define custom search policies driven by external agents. Our implementation is available at https://github.com/xuexue/neuralkanren.

Like the standard miniKanren, this transparent version is implemented in Scheme. To interface with an external agent, we have implemented a Python interface that can drive the miniKanren process via stdin/stdout. Users start by submitting a query, then alternate between receiving constraint tree updates and choosing the next disjunct to unfold.

# 4 Neural Guided Constraint Logic Programming

We present our neural guided synthesis approach summarized in Figure 3. To begin, miniKanren represents the PBE problem in terms of a disjunction of candidate partial programs, and the constraints that must be satisfied for the partial program to be consistent with the examples. A machine learning agent makes discrete choices amongst the possible candidates. The symbolic system then expands the chosen candidate, adding expansions of the candidate to the list of partial programs.

The machine learning model follows these steps:

1. **Embed** the constraints. Sections 4.1 and 4.2 discuss two methods for embedding constraints that trade off ease of training and accounting for logic variable identity.

2. **Score** each constraint. Each constraint embedding is scored independently, using a multi-layer perceptron (MLP).

3. **Pool** scores together. We pool constraint scores for each candidate. We pool hierarchically using the structure of the constraint tree, max-pooling along a disjunction and average-pooling along a conjunction. We find that using average-pooling instead of min-pooling helps gradient flow. In Figure 3 there are no internal disjunctions.

4. **Choose** a candidate. We use a softmax distribution over candidates during training and choose greedily during test.

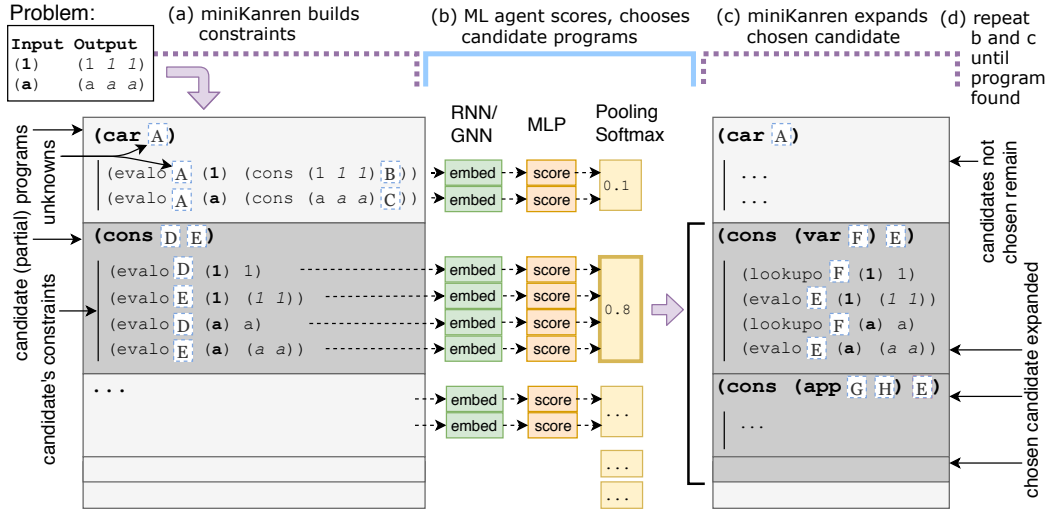

Figure 3: Steps for synthesizing a program that repeats a symbol three times using a subset of Lisp: (a) miniKanren builds constraints representing the PBE problem; candidate programs contain unknowns, whose values are restricted by constraints; (b) a neural agent operating on the constraints scores candidates; each constraint is embedded and scored separately, then pooled per candidate; scores determine which candidate to expand; (c) miniKanren expands the chosen candidate (`cons` D E); possible completions of unknown D are added to the set of candidates; (d) this process continues until a fully-specified program (with no logic variables) is found.

Intuitively, the pooled score for each candidate represents the plausibility of constraints associated with a candidate partial program being satisfied. So in some sense we are learning a neural constraint satisfaction system in order to solve synthesis problems.

## 4.1 Recurrent Neural Network Model (RNN)

One way to embed the constraints is using an RNN operating on each constraint as a sequence. We use an RNN with bi-directional LSTM units [23] to score constraints, with each constraint separately tokenized and embedded. The tokenization process removes identifying information of logic variables, and treats all logic variables as the same token. While logic variable identity is important, since each constraint is embedded and scored separately, the logic variable identity is lost.

We learn separate RNN weights for each relation (`evalo`, `lookupo`, etc). The particular set of constraint types differs depending on the target synthesis language.

## 4.2 Graph Neural Network Model (GNN)

In the RNN model, we lose considerable information by removing the identity of logic variables. Two constraints associated with a logic variable may independently be satisfiable, but may be obviously unsatisfiable together.

To address this, we use a GNN model that embeds all constraints simultaneously. The use of graph or tree structure to represent programs [24, 25] and constraints [26] is not unprecedented. An example graph structure is shown in Figure 4. Each constraint is represented as a tree, but since logic variable leaf nodes may be shared by multiple constraints, the constraint graph is in general a Directed Acyclic Graph (DAG). We do not include the constraint tree structure (disjunctions and conjunctions) in the graph structure since they are handled during pooling.

The specific type of GNN model we use is a Gated Graph Neural Network (GGNN) [27]. Each node has an initial embedding, which is refined through message passing along the edges. The final root node embedding of each constraint is taken to be the embedding representation of the constraint. Since the graph structure is a DAG, we use a synchronous message schedule for message passing.

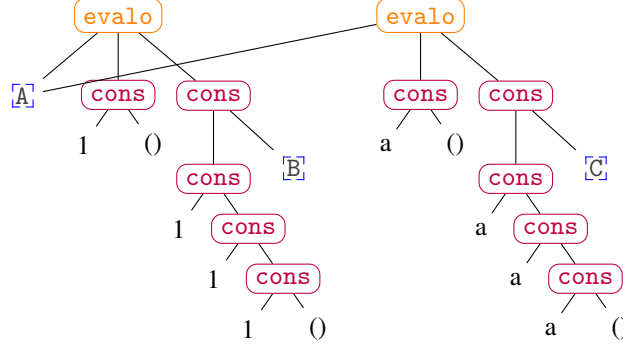

Figure 4: Graph representation of constraints (evalo $\underline{A}$ (1) (cons (1 1 1) $\underline{B}$)) and (evalo $\underline{A}$ (a) (cons (a a a) $\underline{C}$))

One difference between our algorithm and a typical GGNN is the use of different node types. Each token in the constraint tree (e.g. evalo, cons, logic variable) has its own aggregation function and Gated Recurrent Unit weights. Further, the edge types will also follow the node type of the parent node. Most node types will have asymmetric children, so the edge type will also depend on the position of the child.

To summarize, the GNN model has the following steps:

1. **Initialization** of each node, depending on the node type and label. The initial embeddings $e_{label}$ are learned parameters of the model.

2. **Upward Pass**, which is ordered leaf-to-root, so that a node receives all messages from its children and updates its embedding before sending a message to its parents. Since a non-leaf node always has a fixed number of children, the merge function is parameterized as a multi-layer perceptron (MLP) with a fixed size input.

3. **Downward Pass**, which is ordered root-to-leaf, so that a node receives all messages from its parents and updates its embedding before sending a message to its children. Nodes that are not logic variables will only have one parent, so no merge function is required. Constant embeddings are never updated. Logic variables can have multiple parents, so an average pooling is used as a merge function.

4. **Repeat**. The number of upward/downward passes is a hyperparameter. We end on an upward pass so that logic variable updates are reflected in the root node embeddings.

We extract the final embedding of the constraint root nodes for scoring, pooling, and choosing.

## 4.3 Training

We note the similarity in the setup to a Reinforcement Learning problem. The candidates can be thought of as possible *actions*, the ML model as the *policy*, and miniKanren as the non-differentiable *environment* which produces the *states* or constraints. However, during training we have access to the ground-truth optimal action at each step, and therefore use a supervised cross-entropy loss.

We do use other techniques from the Reinforcement Learning literature. We use curriculum learning, beginning with simpler training problems. We generate training states by using the current model parameters to make action choices at least some of the time. We use scheduled sampling [28] with a linear schedule, to increase exploration and reduce teacher-forcing as training progresses. We use prioritized experience replay [29] to reduce correlation in a minibatch, and re-sample more difficult states. To prevent an exploring agent from becoming "stuck", we abort episodes after 20 consecutive incorrect choices. For optimization we use RMSProp [30], with weight decay for regularization.

Importantly, we choose to expand two candidates per step during training, instead of the single candidate as described earlier. We find that expanding two candidates during training allows a better balance of exploration / exploitation during training, leading to a more robust model. During test time, we resume expanding one candidate per step, and use a greedy policy.

# 5 Experiments

Following the programming languages community, we focus on tree manipulation as a natural starting point towards expressive computation. We use a small subset of Lisp as our target language. This subset consists of `cons`, `car`, `cdr`, along with several constants and function application. The full grammar is shown in Figure 5.

```
datum (D)          ::= () | #t | #f | 0 | 1 | x | y | a | b | s | (D . D)
variable-name (V) ::= () | (s . V)
expression (E)     ::= (var V) | (app E E) | (lambda E) | (quote D)
                      | (cons E E) | (car E) | (cdr E) | (list E ...)
```

Figure 5: Subset of Lisp used in this work

We present two experiments. First, we test on programmatically generated synthesis problems held out from training. We compare two miniKanren search strategies that do not use a neural guide, three of our neural-guided models, and RobustFill with a generous beam size. Then, we test the generalizability of these approaches on three families of synthesis problems. In this second set of experiments we additionally compare against state-of-the-art systems $\lambda^2$, Escher, and Myth. All test experiments are run on Intel i7-6700 3.40GHz CPU with 16GB RAM.

## 5.1 Tree Manipulation Problems

We programmatically generate training data by querying (`evalo` $\boxed{P}$ $\boxed{I}$ $\boxed{O}$) in miniKanren, where the program, inputs, and outputs are all unknown. We put several other restrictions on the inputs and outputs so that the examples are sufficiently expressive. When input/output expressions contain constants, we choose random constants to ensure variety. We use 500 generated problems for training, each with 5 input/output examples. In this section, we report results on 100 generated test problems. We report results for several symbolic and neural guided models. Sample generated problems are included in Supplementary Material B.

We compare two variants of symbolic methods that use miniKanren. The "Naive" model uses biased-interleaving search, as described in [31]. The "+ Heuristic" model uses additional hand tuned heuristics described in [16]. The neural guided models include the RNN+Constraints guided search described in Section 4.1 and the GNN+Constraints guided search in Section 4.2. The RNN model uses 2-layer bi-directional LSTMs with embedding size of 128. The GNN model uses a single up/down/up pass with embedding size 64 and message size 128. Increasing the number of passes did not yield improvements. Further, we compare against a baseline RNN model that does not take constraints as input: instead, it computes embeddings of the input, output, and the candidate partial program using an LSTM, then scores the concatenated embeddings using a MLP. This baseline model also uses 2-layer bi-directional LSTMs with embedding size of 128. All models use a 2-layer neural network with ReLU activation as the scoring function.

Table 2 reports the percentage of problems solved within 200 steps. The maximum time the RNN-Guided search used was 11 minutes, so we allow the symbolic models up to 30 minutes. The GNN-Guided search is significantly more computationally expensive, and the RNN baseline model (without constraints) is comparable to the RNN-Guided models (with constraints as inputs).

Table 2: Synthesis Results on Tree Manipulation Problems

| Method | Percent Solved | Average Steps |
|---|---|---|
| Naive [31] | 27% | N/A |
| +Heuristics (Barliman) [16] | 82% | N/A |
| RNN-Guided (No Constraints) | 93% | 46.7 |
| GNN-Guided + Constraints | 88% | 44.5 |
| RNN-Guided + Constraints | 99% | **37.0** |
| RobustFill [4] beam 1000+ | **100%** | N/A |

All three neural guided models performed better than symbolic methods in our tests, with the RNN+Constraints model solving all but one problem. The RNN model without constraints also

performed reasonably, but took more steps on average than other models. RobustFill [4] Attention-C with large beam size solves one more problem than RNN+Constraints on a flattened representation of these problems. Exploration of beam size is in Supplementary Material D. We defer comparison with other symbolic systems because problems in this section involve dynamically-typed, improper list construction.

## 5.2 Generalizability

In this experiment, we explore generalizability. We use the same model weights as above to synthesize three families of programs of varying complexity: `Repeat(N)` which repeats a token $N$ times, `DropLast(N)` which drops the last element in an $N$ element list, and `BringToFront(N)` which brings the last element to the front in an $N$ element list. As a measure of how synthesis difficulty increases with $N$, `Repeat(N)` takes $4 + 3N$ steps, `DropLast(N)` takes $\frac{1}{2}N^2 + \frac{5}{2}N + 1$ steps, and `BringToFront(N)` takes $\frac{1}{2}N^2 + \frac{7}{2}N + 4$ steps. The largest training program takes optimally 22 steps to synthesize. The number of optimal steps in synthesis correlates linearly with program size.

We compare against state-of-the-art systems $\lambda^2$, Escher, and Myth. It is difficult to compare our models against other systems fairly, since these symbolic systems use type information, which provides an advantage. Further, $\lambda^2$ assumes advanced language constructs like `fold` that other methods do not. Escher is built as an active learner, and requires an "oracle" to provide outputs for additional inputs. We do not enable this functionality of Escher, and limit the number of input/output examples to 5 for all methods. We allow every method up to 30 minutes. We also compare against RobustFill Attention-C with a beam size of 5000, the largest beam size supported by our test hardware. Our model is further restricted to 200 steps for consistency with Section 5.1.

Note that if given the full 30 minutes, the RNN+Constraints model is able to synthesize `DropLast(7)` and `BringToFront(6)`, and the GNN+Constraints model is also able to synthesize `DropLast(7)`. Myth solves `Repeat(N)` much faster than our model, taking less than 15ms per problem, but fails on `DropLast` and `BringToFront`. Results are shown in Table 3.

In summary, the RNN+Constraints and GNN+Constraints models both solve problems much larger than those seen in training. The results suggest that using constraints helps generalization: though RobustFill performs best in Section 5.1, it does not generalize to larger problems out of distribution; though RNN+Constraints and RNN-without-constraints perform comparably in Section 5.1, the former shows better generalizability. This is consistent with the observation that as program sizes grow, the corresponding constraints grow more slowly.

Table 3: Generalization Results: largest $N$ for which synthesis succeeded, and failure modes (out of **time**, out of **memory**, requires **oracle**, other **error**)

| Method | Repeat(N) | DropLast(N) | BringToFront(N) |
|---|---|---|---|
| Naive [31] | 6 (time) | 2 (time) | - (time) |
| +Heuristics [16] | 11 (time) | 3 (time) | - (time) |
| RNN-Guided + Constraints | **20+** | **6** (time) | 5 (time) |
| GNN-Guided + Constraints | **20+** | **6** (time) | **6** (time) |
| RNN-Guided (no constraints) | 9 (time) | 3 (time) | 2 (time) |
| $\lambda^2$ [11] | 4 (memory) | 3 (error) | 3 (error) |
| Escher [12] | 10 (error) | 1 (oracle) | - (oracle) |
| Myth [13] | **20+** | - (error) | - (error) |
| RobustFill [4] beam 1000 | 1 | 1 | - (error) |
| RobustFill [4] beam 5000 | 3 | 1 | - (error) |

## 6 Conclusion

We have built a neural guided synthesis model that works directly with miniKanren's constraint representations, and a transparent implementation of miniKanren available at https://github.com/xuexue/neuralkanren. We have demonstrated the success of our approach on challenging tree manipulation and, more importantly, generalization tasks. These results indicate that our approach is a promising stepping stone towards more general computation.

**Acknowledgments**

Research reported in this publication was supported in part by the Natural Sciences and Engineering Research Council of Canada, and the National Center For Advancing Translational Sciences of the National Institutes of Health under Award Number OT2TR002517. R.L. was supported by Connaught International Scholarship. The content is solely the responsibility of the authors and does not necessarily represent the official views of the funding agencies.

## Footnotes

[1]The name "Kanren" comes from the Japanese word for "relation".

[2]In miniKanren convention, a relation is named after the corresponding function, with an 'o' at the end. Supplementary Material A provides a definition of `evalo` used in our experiments.

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
