[Supplementary Material]

# Supplementary Material: Neural Guided Constraint Logic Programming for Program Synthesis

## A   Relational Interpreter

We include below the code for the relational interpreter, written in miniKanren. For readability by machine learning audience, our main paper renames the inputs to the relational interpreter: expr or expression is called P or *program* in the main paper, env or environment is called I or *input*, and value is called O or *output*.

```
(define-relation (evalo expr env value)
  (conde              ;; conde creates a disjunction
    ((fresh (body)    ;; fresh creates new variables and a conjunction
       (== `(lambda ,body) expr)               ;; expr is a lambda definition
       (== `(closure ,body ,env) value)))
    ((== `(quote ,value) expr))                ;; expr is a literal constant
    ((fresh (a*)
       (== `(list . ,a*) expr)                 ;; expr is a list construction
       (eval-listo a* env value)))
    ((fresh (index)
       (== `(var ,index) expr)                 ;; expr is a variable
       (lookupo index env value)))
    ((fresh (rator rand arg env^ body)
       (== `(app ,rator ,rand) expr)           ;; expr is a function application
       (evalo rator env `(closure ,body ,env^))
       (evalo rand env arg)
       (evalo body `(,arg . ,env^) value)))
    ((fresh (a d va vd)
       (== `(cons ,a ,d) expr)                 ;; expr is a cons operation
       (== `(,va . ,vd) value)
       (evalo a env va)
       (evalo d env vd)))
    ((fresh (c vd)
       (== `(car ,c) expr)                     ;; expr is a car operation
       (evalo c env `(,value . ,vd))))
    ((fresh (c va)
       (== `(cdr ,c) expr)                     ;; expr is a cdr operation
       (evalo c env `(,va . ,value)))))))
```

# B Example Generated Problems

Some examples of automatically generated problems are shown in Table A1. Variables in a function body are encoded using de Bruijn indices, so that (var ()) is looking up the 0th (and only) variable. The symbol . denotes a pair.

Table A1: Sample auto-generated training problems

| Program: (LAMBDA (CAR (CAR (VAR ()))))| |
|---|---|
| Input | Output |
| ((b . #t)) | b |
| ((() . b) . a) | () |
| ((a . s) . 1) | a |
| (((y . 1)) . 1) | (y . 1) |
| ((b)) | b |
| **Program: (LAMBDA (CONS (CAR (VAR ())) (QUOTE X)))** | |
| Input | Output |
| (a) | (a . x) |
| (#t . s) | (#t . x) |
| ((1 . y) . y) | ((1 . y) . x) |
| ((y 1 . s) . 1) | ((y 1 . s) . x) |
| (((x . x)) . y) | (((x . x)) . x) |
| **Program: (LAMBDA (QUOTE X))** | |
| Input | Output |
| y | x |
| () | x |
| #t | x |
| a | x |
| b | x |
| **Program: (LAMBDA (CONS (CAR (VAR ())) (CAR (CAR (CDR (VAR ()))))))** | |
| Input | Output |
| (y (y . b) . y) | (y . y) |
| (x (1 . 1)) | (x . 1) |
| (x ((y . a) . x) . a) | (x y . a) |
| ((#f . #t) (#f . a) . 1) | ((#f . #t) . #f) |
| (a ((y #f . #f) . 1) . a) | (a y #f . #f) |
| **Program: (LAMBDA (CAR (CDR (CAR (CAR (CDR (CDR (CDR (VAR ())))))))))** | |
| Input | Output |
| (#f a () ((#f b . 1) . y) . #t) | b |
| (x #t y ((() (#t . a) . s))) | (#t . a) |
| (x b s ((#f (s 1 . b) . y)) . s) | (s 1 . b) |
| (b () #f ((b ((x . #t) . x))) . a) | ((x . #t) . x) |
| (1 #t a ((s (1 #t s . a) . x) . #t) . #t) | (1 #t s . a) |

## C Problems where Neural Guided Synthesis Fails

Table A2 lists problems on which the methods failed. The single problem that RNN + Constraints failed to solve is a fairly complex problem. The problems that the GNN + Constraints failed to solve all include a complex list accessor portion. This actually makes sense: it is conceivable for multi-layer RNNs to be better at this kind of problem compared to a single-layer GNN. The RNN without constraints also fails at complex list accessor problems.

Table A2: Problems where Neural Guided Synthesis Fails

| Method | Problem |
|---|---|
| RNN + Constraints | (lambda (cons (cons (var ()) (var ())) (cons (var ()) (car (cdr (var ())))))) |
| GNN + Constraints | (lambda (car (car (car (car (cdr (cdr (car (var ())))))))))) |
|  | (lambda (car (car (car (cdr (car (cdr (car (var ()))))))))))) |
|  | (lambda (car (car (car (cdr (cdr (cdr (car (var ()))))))))))) |
|  | (lambda (car (car (cdr (car (car (var ())))))))) |
|  | (lambda (car (car (cdr (car (cdr (cdr (car (var ()))))))))))) |
|  | (lambda (car (car (cdr (cdr (cdr (cdr (car (var ()))))))))))) |
|  | (lambda (car (cdr (car (car (cdr (var ())))))))) |
|  | (lambda (car (cdr (car (cdr (cdr (car (var ())))))))))) |
|  | (lambda (car (cdr (cdr (car (car (cdr (var ())))))))))) |
|  | (lambda (car (cdr (cdr (cdr (car (car (var ()))))))))))) |
|  | (lambda (car (cdr (cdr (cdr (car (cdr (var ()))))))))))) |
|  | (lambda (cdr (cdr (car (car (var ()))))))) |
| RNN (No Constraints) | (lambda (cons (car (var ())) (cons (var ()) (cdr (car (var ())))))) |
|  | (lambda (cdr (car (car (cdr (car (cdr (var ())))))))))) |
|  | (lambda (cdr (car (cdr (car (car (car (var ())))))))))) |
|  | (lambda (cdr (car (car (car (car (car (var ())))))))))) |
|  | (lambda (cdr (car (car (cdr (car (cdr (var ())))))))))) |
|  | (lambda (cdr (car (cdr (car (car (car (var ())))))))))) |
|  | (lambda (cdr (car (car (car (car (car (var ())))))))))) |

# D   RobustFill Results for Various Beam Sizes

To compare against RobustFill, we use a flattened representation of the problems shown in Section B, and use the Attention-C model with various beam sizes. For a beam size $k$, if any of the top-$k$ generated programs are correct, we consider the synthesis a success. We report several figures in Table A3: column (a) shows the percent of test problems held out from training that were successfully solved (Table 2 in our paper), and column (b) shows the largest $N$ for a family of synthesis problems for which synthesis succeeds (Table 3 in our paper).

Table A3: RobustFill Results

| Model | (a) Test | (b) Generalization | | |
|---|---|---|---|---|
| | % Solved | Repeat(N) | DropLast(N) | BringToFront(N) |
| RobustFill, Beam Size 1 | 56% | 0 | 0 | 0 |
| RobustFill, Beam Size 10 | 94% | 0 | 0 | 0 |
| RobustFill, Beam Size 100 | 99% | 1 | 0 | 0 |
| RobustFill, Beam Size 1000 | 100% | 1 | 1 | 0 |
| RobustFill, Beam Size 5000 | 100% | 3 | 1 | 0 |
| RNN-Guided + Constraints (Ours) | 99% | 20+ | 6 | 5 |