[Reviews · NeurIPS 2018]

Reviewer 1



This paper shows that exposing the constraints in an AND-OR tree to a neural-trained search heuristic can make the search succeed sooner and generalize better. In particular, the paper uses constraint logic programming language miniKanren, guided by such a heuristic to decide which disjunct to explore next, to synthesize simple tree-manipulation programs. The idea is valuable and the paper explains it well. I am disappointed that the language is so simple as to not include conditionals. Of course, conditionals can in principle be encoded using lambda and app, but the tree-manipulation programs synthesized do not perform tasks that call for conditionals, such as feeding every element of a list to a given function. A more adequate evaluation might take advantage of other tasks that existing systems such as [11-13] aim for. After all, the paper says "we are interested in exploring manipulations of richer data structures." It is inaccurate to say that to "focus on list construction" is "following the programming language community". The discussion of related work is limited to program synthesis, and should be expanded to include other uses of neural-trained heuristics in search. The discussion in Section 4 about whether and how to embed logic variable identity in a neural representation is interesting. Thus, claims such as "the logic variable identity is of limited use" and "we lose considerable information by removing the identity of logic variables" should be supported empirically, such as by ablation experiments. On page 4, "variable identify" -> "variable identity" On page 6, please elaborate on what you mean by "a supervised cross-entropy loss". The next paragraph can also use elaboration. I read and appreciate the author response. Regarding "a supervised cross-entropy loss", I'm still not completely clear -- what does "ground truth step" mean (in particular, what is its type) in "L = − log p(ground truth step)"?

Reviewer 2



This paper presents a neural-guided approach for input-output program synthesis problems, which is a trending problem in the neural program synthesis community. The main idea is to incorporate constraint logic programming during synthesis, and employ neural networks to rank the different constraints during the unfold procedure when searching for a desired program satifying input-output examples. The proposed approach is sound to me. I'm in between 5 and 6. My main concern is that the proposed approach is mainly compared against previous works in the PL community, and may neglect several advancements from the neural program synthesis literature in recent years. For example, [1] deals a similar program space using a sampling approach. [2, 3] provides the neural program synthesis approaches for the FlashFill task, which is quite similar to the string manipulation task evaluated in this paper. [4] proposes a reinforcement learning-based approach to guide synthesizing a mixture of differentiable and non-differentiable programs, which is related to the approach of this work, though might not be directly comparable. After reading the rebuttal, I think it contains decent results, which should be included in the final version. So I raise my rating. [1] Ellis et al. Sampling for Bayesian Program Learning. NIPS 2016 [2] Delvin et al. Robustfill: Neural program learning under noisy I/O. ICML 2017 [3] Parisotto et al. Neuro-symbolic program synthesis. ICLR 2017 [4] Chen et al. Toward Synthesizing Complex Programs from Input-Output Examples. ICLR 2018

Reviewer 3



The paper presents an approach to neural guided search for program synthesis of LISP programs. A modified miniKanren constraint solver is used to synthesize a program from example IO pairs by "inverting" the EVAL function. The search for candidate programs in the solver is guided by a neural network (either a GNN or an RNN with other layers on top). The approach is compared to baselines and three symbolic methods. Pros: - The paper is very well written and definitely relevant to NIPS - The evaluation against symbolic methods is reasonable - While the idea of scoring evaluation expansions is not novel, applying it to the EVAL function is Cons: - Novelty is somewhat limited - Links to some recent related work are missing - No comparison to other neural methods In general, I think that paper is clear and sound enough. I am not so confortable about originality, but still, I am in favor of acceptance. Major issues: - The fact that learners can be used to score candidate rule expansions has already been demonstrated (several times) before, see for instance: Learning to Search in Branch-and-Bound Algorithms, 2014, He, Daume' III, and Eisner (more distantly related; but note that, unsurprisingly, program synthesis and structure learning in general *can* be framed as a MILP problem) End-to-End Differentiable Proving, 2017, Rocktäschel and Riedel (in a very related setting) Essentially this paper "ports" the neural guidance idea from SLD resolution to constraint satisfaction, which in miniKanren is based on the same principle. The handling of logical operators as aggregators is also similar. As far as I can see, the novely here is in the choice of "what to guide", namely the expansion of the EVAL function, and in the choice of neural architecture. This is less novel than the authors seem to claim. This is good enough, but the authors should definitely better position their contribution w.r.t. the aforementioned works, especially the second one. - The fact that the proposed system is "deeply integrated" with miniKanren and LISP prevents comparison to existing neural approaches. This is very unfortunate---but unavoidable, I guess. More generally, it is difficult to see what the advantages of the proposed approach are with respect to existing neural synthesis alternatives. Unfortunately there is no ready-made, agreed-upon benchmark or problem setting for neural program synthesis, and I understand that it is difficult to compare against other neural techniques. However, this is only a partial excuse. Note that, for instance, since in principle prolog can be used to synthesize programs satisfying a set of examples, the method in the "end-to-end differentiable proving" paper can be used for program synthesis too. I won't ask the authors to compare against it, as it is a lot of work, but this should be doable. - The pros and cons of RNN versus GNN are not really studied in depth. I guess that this is unavoidable, given that these models are quite opaque. Minor issues: - missing commas in Figure 2